# BI-TUNING OF PRE-TRAINED REPRESENTATIONS

## ABSTRACT

It is common within the deep learning community to first *pre-train* a deep neural network from a large-scale dataset and then *fine-tune* the pre-trained model to a specific downstream task. Recently, both supervised and unsupervised pre-training approaches to learning representations have achieved remarkable advances, which exploit the discriminative knowledge of labels and the intrinsic structure of data, respectively. It follows natural intuition that both discriminative knowledge and intrinsic structure of the downstream task can be useful for fine-tuning, however, existing fine-tuning methods mainly leverage the former and discard the latter. A question arises: How to fully explore the intrinsic structure of data for boosting fine-tuning? In this paper, we propose Bi-tuning, a general learning approach to fine-tuning both supervised and unsupervised pre-trained representations to downstream tasks. Bi-tuning generalizes the vanilla fine-tuning by integrating two heads upon the backbone of pre-trained representations: a classifier head with an improved contrastive cross-entropy loss to better leverage the label information in an instance-contrast way, and a projector head with a newly-designed categorical contrastive learning loss to fully exploit the intrinsic structure of data in a category-consistent way. Comprehensive experiments confirm that Bi-tuning achieves state-of-the-art results for fine-tuning tasks of both supervised and unsupervised pre-trained models by large margins (*e.g.* 10.7% absolute rise in accuracy on *CUB* in low-data regime).

## 1 INTRODUCTION

In the last decade, remarkable advances in deep learning have been witnessed in diverse applications across many fields, such as computer vision, robotic control, and natural language processing in the presence of large-scale labeled datasets. However, in many practical scenarios, we may have only access to a small labeled dataset, making it impossible to train deep neural networks from scratch. Therefore, it has become increasingly common within the deep learning community to first *pre-train* a deep neural network from a large-scale dataset and then *fine-tune* the pre-trained model to a specific downstream task. Fine-tuning requires fewer labeled data, enables faster training, and usually achieves better performance than training from scratch (He et al., 2019). This two-stage style of pre-training and fine-tuning lays as the foundation of various transfer learning applications.

In the *pre-training* stage, there are mainly two approaches to pre-train a deep model: supervised pre-training and unsupervised pre-training. Recent years have witnessed the success of numerous supervised pre-trained models, *e.g.* ResNet (He et al., 2016), by exploiting the discriminative knowledge of labels on a large-scale dataset like ImageNet (Deng et al., 2009). Meanwhile, unsupervised representation learning is recently changing the field of NLP by models pre-trained with a large-scale corpus, *e.g.* BERT (Devlin et al., 2018) and GPT Radford & Sutskever (2018). In computer vision, remarkable advances in unsupervised representation learning (Wu et al., 2018; He et al., 2020; Chen et al., 2020), which exploit the intrinsic structure of data by contrastive learning Hadsell et al. (2006), are also to change the field dominated chronically by supervised pre-trained representations.

In the *fine-tuning* stage, transferring a model from supervised pre-trained models has been empirically studied in Kornblith et al. (2019). During the past years, several sophisticated fine-tuning methods were proposed, including L2-SP (Li et al., 2018), DELTA (Li et al., 2019) and BSS (Chen et al., 2019). These methods focus on leveraging the discriminative knowledge of labels by a cross-entropy loss and the implicit bias of pre-trained models by a regularization term. However, the intrinsic structure of data in downstream task is generally discarded during fine-tuning. Further, we empirically observed that unsupervised pre-trained representations focus more on the intrinsic structure, while supervised

pre-trained representations explain better on the label information (Figure 3). This possibly implies that fine-tuning unsupervised pre-training representations is may be more difficult He et al. (2020).

Regarding to the success of supervised and unsupervised pre-training approaches, it follows a natural intuition that both *discriminative knowledge* and *intrinsic structure* of the downstream task can be useful for fine-tuning. A question arises: How to fully explore the intrinsic structure of data for boosting fine-tuning? To tackle this major challenge of deep learning, we propose **Bi-tuning**, a general learning approach to fine-tuning both supervised and unsupervised pre-trained representations to downstream tasks. Bi-tuning generalizes the vanilla fine-tuning by integrating two heads upon the backbone of pre-trained representations:

- A classifier head with an improved contrastive cross-entropy loss to better leverage the label information in an instance-contrast way, which is the dual view of the vanilla cross-entropy loss and is expected to achieve a more compact intra-class structure.

- A projector head with a newly-designed categorical contrastive learning loss to fully exploit the intrinsic structure of data in a category-consistent way, resulting in a more harmonious cooperation between the supervised and unsupervised fine-tuning mechanisms.

As a general fine-tuning approach, Bi-tuning can be applied with a variety of backbones without any additional assumptions. Comprehensive experiments confirm that Bi-tuning achieves state-of-the-art results for fine-tuning tasks of both supervised and unsupervised pre-trained models by large margins (*e.g.* 10.7% absolute rise in accuracy on *CUB* in low-data regime). We justify through ablation studies the effectiveness of the proposed two-heads fine-tuning architecture with their novel loss functions.

## 2 RELATED WORK

### 2.1 PRE-TRAINING

During the past years, supervised pre-trained models achieve impressive advances by exploiting the inductive bias of label information on a large-scale dataset like ImageNet (Deng et al., 2009), such as GoogleNet (Szegedy et al., 2015), ResNet (He et al., 2016), DenseNet (Huang et al., 2017), to name a few. Meanwhile, unsupervised representation learning is recently shining in the field of NLP by models pre-trained with a large-scale corpus, including GPT (Radford & Sutskever, 2018), BERT (Devlin et al., 2018) and XLNet (Yang et al., 2019). Even in computer vision, recent impressive advances in unsupervised representation learning (Wu et al., 2018; He et al., 2020; Chen et al., 2020), which exploit the inductive bias of data structure, are shaking the long-term dominated status of representations learned in a supervised way. Further, a wide range of handcrafted pretext tasks have been proposed for unsupervised representation learning, such as relative patch prediction (Doersch et al., 2015), solving jigsaw puzzles (Noroozi & Favaro, 2016), colorization (Zhang et al., 2016), etc.

### 2.2 CONTRASTIVE LEARNING

Specifically, various unsupervised pretext tasks are based on some forms of contrastive learning, in which the instance discrimination approach (Wu et al., 2018; He et al., 2020; Chen et al., 2020) is one of the most general forms. It is noteworthy that the spirits of contrastive learning actually can date back very far (Becker & Hinton, 1992; Hadsell et al., 2006; Gutmann & Hyvärinen, 2010). The key idea is to maximize the likelihood of the distribution $p(\mathbf{x}|D)$ contrasting to the artificial noise distribution $p_n(\mathbf{x})$, also known as noise-contrastive estimation (NCE). Later, Goodfellow et al. (2014) pointed out the relations between generative adversarial networks and noise-contrastive estimation. Meanwhile, (van den Oord et al., 2018) revealed that contrastive learning is related to mutual information between a query and the corresponding positive key, which is known as InfoNCE.

Other variants of contrastive learning methods include contrastive predictive learning (CPC) (van den Oord et al., 2018) and colorization contrasting (Tian et al., 2019). Recent advances of deep contrastive learning benefit from contrasting positive keys against *very large* number of negative keys. Therefore, how to efficiently generate keys becomes a fundamental problem in contrastive learning. To achieve this goal, Doersch & Zisserman (2017) explored the effectiveness of in-batch samples, Wu et al. (2018) proposed to use a memory bank to store all representations of the dataset, He et al. (2020)

further replaced a memory bank with the momentum contrast (MoCo) to be memory-efficient, and Chen et al. (2020) showed that a brute-force huge batch of keys works well.

## 2.3 FINE-TUNING

Fine-tuning a model from supervised pre-trained models has been empirically explored in Kornblith et al. (2019) by launching a systematic investigation with grid search of the hyper-parameters. During the past years, a few fine-tuning methods have been proposed to exploit the inductive bias of pre-trained models: L2-SP (Li et al., 2018) drives the weight parameters of target task to the pre-trained values by imposing L2 constraint based on the inductive bias of parameter; DELTA (Li et al., 2019) computes channel-wise discriminative knowledge to reweight the feature map regularization with an attention mechanism based on the inductive bias of behavior; BSS (Chen et al., 2019) penalizes smaller singular values to suppress untransferable spectral components based on singular values.

Other fine-tuning methods including learning with similarity preserving (Kang et al., 2019) or learning without forgetting (Li & Hoiem, 2017) also work well on some downstream classification tasks. However, the existing fine-tuning methods mainly focus on leveraging the knowledge of the target label with a cross-entropy loss. Intuitively, encouraging a model to capture the label information and intrinsic structure simultaneously may help the model transition between the upstream unsupervised models with the downstream classification tasks. In natural language processing, GPT (Radford & Sutskever, 2018; Radford et al., 2019) has employed a strategy that jointly optimizes unsupervised training criteria while fine-tuning with supervision. *However, we empirically found that trivially following this kind of force-combination between supervised learning loss and unsupervised contrastive learning loss is beneficial but limited.* The plausible reason is that these two losses will contradict with each other and result in an extremely different but not discriminative feature structure compared to that of the supervised cross-entropy loss (See Figure 3).

## 3 CONTRASTIVE LEARNING WITH MULTIPLE POSITIVE KEYS

Instance discrimination approach (van den Oord et al., 2018; Wu et al., 2018), *a.k.a.* InfoNCE, is one of the most general forms of standard contrastive learning. Given a query $\mathbf{q}$ with a large key pool $\{\mathbf{k}_0, \mathbf{k}_1, \mathbf{k}_2, \cdots, \mathbf{k}_K\}$ where $K$ is the number of keys, this non-parametric contrastive loss is

$$L_{\text{InfoNCE}} = -\log \frac{\exp(\mathbf{q} \cdot \mathbf{k}_+/\tau)}{\sum_{i=0}^{K} \exp(\mathbf{q} \cdot \mathbf{k}_i/\tau)}, \tag{1}$$

where $\tau$ is the hyper-parameter for temperature scaling. Intuitively, contrastive learning can be defined as a query-key pair matching problem, in which a contrastive loss is a $K$-way cross-entropy loss to distinguish $\mathbf{k}_+$ from a large key pool. From this perspective, a contrastive loss is to maximize the similarity between the query and the corresponding positive key $\mathbf{k}_+$.

When fine-tuning the pre-trained model on a labeled downstream dataset, an intuitive way is to combine InfoNCE and cross-entropy as $\mathcal{L} = L_{\text{InfoNCE}} + L_{\text{CE}}$. However, InfoNCE tends to generate an extremely different but not discriminative feature structure compared with that of the supervised cross-entropy loss, making the classifier struggle. To this end, we proposed a novel idea of *contrastive learning with multiple positive keys* to tailor contrast into cross-entropy loss in both the projector head and classifier head. Before describing the losses defined on these two heads in detail, let us make a brief introduction of contrastive learning with multiple positive keys.

**Definition 1 (Contrastive Learning with Multiple Positive Keys)** *In the context of tailoring contrastive into fine-tuning the pre-trained model on a labeled downstream task, it is the mechanism that expands the scope of positive keys to a set of instances instead of a single one.*

Formally, given a query $\mathbf{q}$ with a large key pool $\{\mathbf{k}_0, \mathbf{k}_1, \mathbf{k}_2, \cdots, \mathbf{k}_K\}$ where $K$ is the number of keys and $\mathbf{k}_0$ is the positive key $\mathbf{k}_+$. Suppose the intance number in each class is equal in the key pool, *i.e.*, the equality $K = k \cdot C$ held where $C$ is the size of label space and $k$ is the instance number in each class. The straightforward approach to expand the standard contrastive loss ($L_{\text{InfoNCE}}$) into a form of multi-positives (denoted by a positive key set $\mathbf{K}_p$) is

$$L'_{\text{InfoNCE}} = -\frac{1}{|\mathbf{K}_p|} \sum_{\mathbf{k}_+ \in \mathbf{K}_p} \log \frac{\exp(\mathbf{q} \cdot \mathbf{k}_+/\tau)}{\exp(\mathbf{q} \cdot \mathbf{k}_+/\tau) + \sum_{\mathbf{k}_- \in \mathbf{K}_n} \exp(\mathbf{q} \cdot \mathbf{k}_-/\tau)}, \tag{2}$$

where $\mathbf{K}_n$ denotes the negative key set. We denote this loss function by $L'_{\text{InfoNCE}}$, which essentially performs multiple *individual* contrasts with different positive keys for each query $q$. However, this form of multi positive key contrastive can be seen as a simple generaliztion of the standard contrastive loss by repeating the positve keys several times, without fully exploiting the intrinsic structure of intra-class dataset. To this end, the losses ($L_{\text{CCE}}$ and $L_{\text{CCL}}$ detailed in Section 4.3 and Section 4.4 respectively) we proposed are based on the following formula:

$$L_{\text{proposed}} = -\frac{1}{|\mathbf{K}_p|} \sum_{\mathbf{k}_+ \in \mathbf{K}_p} \log \frac{\exp(\mathbf{q} \cdot \mathbf{k}_+/\tau)}{\sum_{k^+ \in \mathbf{K}_p} \exp(\mathbf{q} \cdot \mathbf{k}_+/\tau) + \sum_{\mathbf{k}_- \in \mathbf{K}_n} \exp(\mathbf{q} \cdot \mathbf{k}_-/\tau)} \tag{3}$$

Different from $L'_{\text{InfoNCE}}$, in the denominator of $L_{\text{proposed}}$, both the positive keys in the same class with the query and the negative keys from other classes are presented. For each contrast with multiple positive keys, a query here needs to *balance* all positive keys simultaneously. In another view, $L_{\text{proposed}}$ can be regarded as performing cross-entropy on soft labels of uniform probability $\frac{1}{|\mathbf{K}_p|}$ for each positive key. $L_{\text{proposed}}$ introduces a *uninformative prior* that the positive keys are uniformly distributed around the query. Table 6 compared the accuracies of Bi-tuning implemented with both forms of multiple-positives and revealed that the proposed form is the better choice.

# 4 METHOD

## 4.1 PRE-TRAINED REPRESENTATIONS

Bi-tuning is a general learning approach to fine-tuning both supervised and unsupervised representations. Without any additional assumptions, the pre-trained feature encoder $f(\cdot)$ can be various network backbones according to the downstream tasks, including ResNet (He et al., 2016) and DenseNet (Huang et al., 2017) for supervised pre-trained models, and MoCo (He et al., 2020) and SimCLR (Chen et al., 2020) for unsupervised pre-trained models. Given a query sample $\mathbf{x}_i^q$, we can first utilize a pre-trained feature encoder $f(\cdot)$ to extract its pre-trained representation as $\mathbf{h}_i^q = f(\mathbf{x}_i^q)$.

## 4.2 VANILLA FINE-TUNING

Given a pre-trained representation $\mathbf{h}_i^q$, a fundamental step of vanilla fine-tuning is to feedforward the representation $\mathbf{h}_i^q$ into a $C$-way classifier $g(\cdot)$, in which $C$ is the number of categories for the downstream classification task. Denote the parameters of the classifier $g(\cdot)$ as $\mathbf{W} = [\mathbf{w}_1, \mathbf{w}_2, \cdots, \mathbf{w}_C]$, where $\mathbf{w}_j$ corresponds to the parameter for the $j$-th class. Denote the training dataset of the downstream task as $\{(\mathbf{x}_i^q, y_i^q)\}_{i=1}^N$. $\mathbf{W}$ can be updated by optimizing a standard cross-entropy (CE) loss as

$$L_{\text{CE}} = -\sum_{i=1}^N \log \frac{\exp(\mathbf{w}_{y_i} \cdot \mathbf{h}_i^q)}{\sum_{j=1}^C \exp(\mathbf{w}_j \cdot \mathbf{h}_i^q)}. \tag{4}$$

## 4.3 CONTRASTIVE CROSS-ENTROPY LOSS ON CLASSIFIER HEAD

From another perspective, the cross-entropy loss of vanilla fine-tuning on a given dataset with $N$ instance-class pair $(\mathbf{x}_i^q, y_i^q)$ can be regarded as a class-wise championship, *i.e.*, the prediction that is the same as the ground-truth class of each instance is expected to win the game. To further exploit the label information of the downstream task, we propose a novel contrastive cross-entropy loss $L_{\text{CCE}}$ on the classifier head via the dual view of cross-entropy loss. Similarly, $L_{\text{CCE}}$ can be seen as an instance-wise championship, *i.e.*, the prediction belongs to the nearest instance towards the prototype of each class is expected to win the game. Similar to CE loss, $L_{\text{CCE}}$ can be formulated as

$$L_{\text{CCE}} = -\frac{1}{|\mathbf{K}_p|} \sum_{h^+ \in \mathbf{K}_p} \log \frac{\exp(\mathbf{w}_y \cdot \mathbf{h}^+/\tau)}{\sum_{h^+ \in \mathbf{K}_p} \exp(\mathbf{w}_y \cdot \mathbf{h}^+/\tau) + \sum_{h^- \in \mathbf{K}_n} \exp(\mathbf{w}_y \cdot \mathbf{h}^-/\tau)}, \tag{5}$$

where $\mathbf{K}_p$ is the positive key set (including the current query example $\mathbf{h}^q$ and keys with the same label $y^q$), and $\mathbf{K}_n$ is the negative set (including examples with other classes). Note that, $\mathbf{h}$'s are samples from the hidden key pool produced by the key generating mechanism (except $\mathbf{h}^q$). Though Bi-tuning is general for it, we adopt the key generating approach in Momentum Contrast (MoCo) (He et al.,

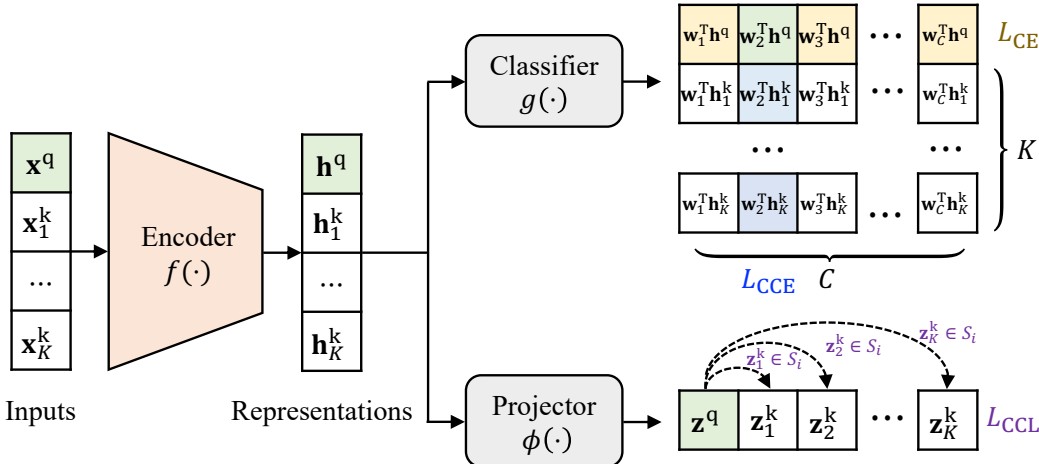

Figure 1: The architecture of the proposed **Bi-tuning** approach, which includes an encoder for pre-trained representations, a classifer head and a projector head. Bi-tuning enables a dual fine-tuning mechanism: a contrastive cross-entropy loss (CCE) on the classifier head to exploit label information and a categorical contrastive learning loss (CCL) on the projector head to model the intrinsic structure.

2020) as our default one due to its simplicity, high-efficacy, and memory-efficient implementation. As intuitively illustrated in Figure 1, column-wised $L_{\text{CCE}}$ operates loss computation along the bank dimension whose size is $K + 1$ while row-wised $L_{\text{CE}}$ performs along the class dimension with a size of $C$. By encouraging instances in the training dataset to approach towards their corresponding class prototypes, $L_{\text{CCE}}$ tends to achieve a more compact intra-class structure than the vanilla fine-tuning.

### 4.4 CATEGORICAL CONTRASTIVE LEARNING LOSS ON PROJECTOR HEAD

Previously, we proposed an improved-version of vanilla fine-tuning on the classifier head to fully exploit label information. However, this kind of loss function design may still fall short in capturing the intrinsic structure. Inspired by the remarkable success of unsupervised pre-training which also aims at modeling the intrinsic structure in data, we first introduce a projector $\phi(\cdot)$ which is usually off the shelf to embed a pre-trained representation $\mathbf{h}_i^{\text{q}}$ into a latent metric space as $\mathbf{z}_i^{\text{q}}$. However, the standard contrastive learning loss (InfoNCE) defined in Eq. (1) assumes that there is a *single* key $k_+$ in the dictionary to match the given query $q$, which implicitly requires every instance to belong to an individual class. If we simply apply InfoNCE loss on the labeled downstream dataset, it will result in an extremely different but not discriminative feature structure compared with that of the supervised cross-entropy loss, making the classifier struggle. Obviously, this dilemma reveals that the naive combination of the supervised cross-entropy loss and the unsupervised contrastive loss is not an optimal solution for fine-tuning, which is also backed by our experiments in Table 3.

To capture the label information and intrinsic structure simultaneously, we propose a novel categorical contrastive loss $L_{\text{CCL}}$ on the projector head based on the following hypothesis: when we fine-tune a pre-trained model to a downstream task, it is reasonable to regard other keys in the same class as the positive keys that the query matches. In this way, $L_{\text{CCL}}$ expands the scope of positive keys to *a set of instances* instead of a *single one*, resulting in a more harmonious cooperation between the supervised and unsupervised learning mechanisms. Similar to the format of InfoNCE loss, $L_{\text{CCL}}$ is defined as

$$L_{\text{CCL}} = -\frac{1}{|\mathbf{K}_p|} \sum_{\mathbf{z}^+ \in \mathbf{K}_p} \log \frac{\exp(\mathbf{z}^{\text{q}} \cdot \mathbf{z}^+/\tau)}{\sum_{\mathbf{z}^+ \in \mathbf{K}_p} \exp(\mathbf{z}^{\text{q}} \cdot \mathbf{z}^+/\tau) + \sum_{\mathbf{z}^- \in \mathbf{K}_n} \exp(\mathbf{z}^{\text{q}} \cdot \mathbf{z}^-/\tau)}, \quad (6)$$

where the notations are identical to Eq. (5), as well as the positive key set. Note that, the outer sum is over all positive keys, indicating that there may be more than one positive key for a single query, *i.e.*, the inequality that $|\mathbf{K}_p| \geq 1$ holds.

Table 1: Top-1 accuracy on various datasets using ResNet-50 by *supervised pre-training*.

| Dataset | Method | Sampling Rates | | | |
| | | 25% | 50% | 75% | 100% |
| --- | --- | --- | --- | --- | --- |
| CUB | Fine-tuning (baseline) | 61.36±0.11 | 73.61±0.23 | 78.49±0.18 | 80.74±0.15 |
| | L2SP (Li et al., 2018) | 61.21±0.19 | 72.99±0.13 | 78.11±0.17 | 80.92±0.22 |
| | DELTA (Li et al., 2019) | 62.89±0.11 | 74.35±0.28 | 79.18±0.24 | 81.33±0.24 |
| | BSS (Chen et al., 2019) | 64.69±0.31 | 74.96±0.21 | 78.91±0.15 | 81.52±0.11 |
| | **Bi-tuning** | **67.47**±0.08 | **77.17**±0.13 | **81.07**±0.09 | **82.93**±0.23 |
| Cars | Fine-tuning (baseline) | 56.45±0.21 | 75.24±0.17 | 83.22±0.17 | 86.22±0.12 |
| | L2SP (Li et al., 2018) | 56.29±0.21 | 75.62±0.32 | 83.60±0.13 | 85.85±0.12 |
| | DELTA (Li et al., 2019) | 58.74±0.23 | 76.53±0.08 | 84.53±0.29 | 86.01±0.37 |
| | BSS (Chen et al., 2019) | 59.74±0.14 | 76.78±0.16 | 85.06±0.13 | 87.64±0.21 |
| | **Bi-tuning** | **66.15**±0.20 | **81.10**±0.07 | **86.07**±0.23 | **88.47**±0.11 |
| Aircraft | Fine-tuning (baseline) | 51.25±0.18 | 67.12±0.41 | 75.22±0.09 | 79.18±0.20 |
| | L2SP (Li et al., 2018) | 51.07±0.45 | 67.46±0.22 | 75.06±0.45 | 79.07±0.21 |
| | DELTA (Li et al., 2019) | 53.71±0.30 | 68.51±0.24 | 76.51±0.55 | 80.34±0.14 |
| | BSS (Chen et al., 2019) | 53.38±0.22 | 69.19±0.18 | 76.39±0.22 | 80.83±0.32 |
| | **Bi-tuning** | **58.27**±0.26 | **72.40**±0.22 | **80.77**±0.10 | **84.01**±0.33 |

### 4.5 BI-TUNING

Finally, we reach a novel approach to fine-tuning both supervised and unsupervised representations, *i.e.* the **Bi-tuning**, which jointly optimizes the standard cross-entropy loss, the contrastive cross-entropy loss for classifier head and the categorical contrastive learning loss for projector head in an end-to-end deep architecture. Note that, Bi-tuning refers to the proposed two heads (*a.k.a*, bi-head) with two novel losses. The overall loss function of Bi-tuning can be formulated as follows:

$$\min_{\Theta} L_{\text{CE}} + L_{\text{CCE}} + L_{\text{CCL}}, \tag{7}$$

where $\Theta$ denotes the set of all parameters of the backbone, the classifier head and the projector head. Specifically, since the magnitude of the above loss terms is comparable, we empirically find that there is no need to introduce any extra hyper-parameters to trade-off them. This simplicity makes Bi-tuning easy to be applied to different datasets or tasks. The full portrait of Bi-tuning is shown in Figure 1.

## 5 EXPERIMENTS

We follow the common fine-tuning principle described in Yosinski et al. (2014), replacing the last task-specific layer in the classifier head with a randomly initialized fully connected layer whose learning rate is 10 times of that for pre-trained parameters. Meanwhile, the projector head is set to be another randomly initialized fully connected layer. For the key generating mechanisms, we follow the style in He et al. (2020), employing a momentum contrast branch with a default momentum coefficient $m = 0.999$ and two cached queues both normalized by their L2-norm (Wu et al., 2018) with dimensions of 2048 and 128 respectively. For each task, the best learning rate is selected by cross-validation under a 100% sampling rate and applied to all four sampling rates. Queue size $K$ is set as $8, 16, 24, 32$ for each category according to the dataset scales respectively. Other hyper-parameters in Bi-tuning are fixed for all experiments. The temperature $\tau$ in Eq. (5) and Eq. (6) is set as 0.07 (Wu et al., 2018). The trade-off coefficients between these three losses are kept as 1 since the magnitude of the loss terms is comparable. All tasks are optimized using SGD with a momentum 0.9. All results in this sections are averaged over 5 trails and standard deviations are provided.

### 5.1 BI-TUNING SUPERVISED PRE-TRAINED REPRESENTATIONS

**Standard benchmarks.** We first verify our approach on three fine-grained classification benchmarks: **CUB-200-2011** (Welinder et al., 2010) (with 11788 images for 200 bird species), **Stanford Cars** (Krause et al., 2013) (containing 16185 images of 196 classes of cars) and **FGVC Aircraft** (Maji et al., 2013) (containing 10000 samples 100 different aircraft variants). For each benchmark,

Table 2: Top-1 accuracy on **COCO-70** dataset using DenseNet-121 by *supervised pre-training*.

| Dataset | Method | Sampling Rates | | | |
|---------|--------|------|------|------|------|
| | | 25% | 50% | 75% | 100% |
| COCO-70 | Fine-tuning (baseline) | 80.01±0.25 | 82.50±0.25 | 83.43±0.18 | 84.41±0.22 |
| | L2SP (Li et al., 2018) | 80.57±0.47 | 80.67±0.29 | 83.71±0.24 | 84.78±0.16 |
| | DELTA (Li et al., 2019) | 76.39±0.37 | 79.72±0.24 | 83.01±0.11 | 84.66±0.08 |
| | BSS (Chen et al., 2019) | 77.29±0.15 | 80.74±0.22 | 83.89±0.09 | 84.71±0.13 |
| | **Bi-tuning** | **80.68**±0.23 | **83.48**±0.13 | **84.16**±0.05 | **85.41**±0.23 |

we create four configurations which randomly sample 25%, 50%, 75%, and 100% of training data for each class respectively, to reveal the detailed effect while fine-tuning to different data scales. We choose recent fine-tuning technologies: L2-SP (Li et al., 2018), DELTA (Li et al., 2019), and the state-of-the-art method BSS (Chen et al., 2019), as competitors of Bi-tuning while regarding vanilla fine-tuning as a baseline. Note that vanilla fine-tuning is a strong baseline when sufficient data is provided. Results are averaged over 5 trials. As shown in Table 1, Bi-tuning significantly outperforms all competitors across all three benchmarks by large margins (*e.g.* 10.7% absolute rise on *CUB* with a sampling rate of 25%). Note that even under 100% sampling rate, Bi-tuning still outperforms others.

**Large-scale benchmarks.** Previous fine-tuning methods mainly focus on improving performance under low-data regime paradigms. We further extend Bi-tuning to large-scale paradigms. We use annotations of COCO dataset (Lin et al., 2014) to construct a large-scale classification dataset, cropping object with padding for each image and removing minimal items (with height and width less than 50 pixels), resulting a large-scale dataset containing 70 classes with more than 1000 images per category. The scale is comparable to ImageNet in terms of the number of samples per class. On this constructed large-scale dataset named COCO-70, Bi-tuning is also evaluated under four sampling rate configurations. Since even 25% sampling rates of COCO-70 is much larger than each benchmark in Section 5.1, previous fine-tuning competitors show micro contributions to these paradigms. Results in Table 2 reveal that Bi-tuning brings general gains for all tasks,beyond the low-data regime. We hypothesize that the intrinsic structure introduced by Bi-tuning contributes substantially.

Table 3: Top-1 accuracy on various datasets using ResNet-50 *unsupervisedly pre-trained* by MoCo.

| Dataset | Method | Sampling Rates | | | |
|---------|--------|------|------|------|------|
| | | 25% | 50% | 75% | 100% |
| CUB | Fine-tuning (baseline) | 38.57±0.13 | 58.97±0.16 | 69.55±0.18 | 74.35±0.18 |
| | GPT* (Radford et al., 2019) | 36.43±0.17 | 57.62±0.14 | 67.82±0.05 | 72.95±0.29 |
| | Center (Wen et al., 2016) | 42.53±0.41 | 62.15±0.51 | 70.86±0.39 | 75.61±0.33 |
| | BSS (Chen et al., 2019) | 41.73±0.14 | 59.15±0.21 | 69.93±0.19 | 74.16±0.09 |
| | **Bi-tuning** | **50.54**±0.23 | **66.88**±0.13 | **74.27**±0.05 | **77.14**±0.23 |
| Cars | Fine-tuning (baseline) | 62.40±0.26 | 81.55±0.36 | 88.07±0.19 | 89.81±0.48 |
| | GPT* (Radford et al., 2019) | 65.83±0.27 | 82.39±0.17 | 88.62±0.11 | 90.56±0.18 |
| | Center (Wen et al., 2016) | 67.57±0.12 | 82.78±0.30 | 88.55±0.24 | 89.95±0.1 |
| | BSS (Chen et al., 2019) | 62.13±0.22 | 81.72±0.22 | 88.32±0.17 | 90.41±0.15 |
| | **Bi-tuning** | **69.44**±0.32 | **84.41**±0.07 | **89.32**±0.23 | **90.88**±0.13 |
| Aircraft | Fine-tuning (baseline) | 58.98±0.54 | 77.39±0.31 | 84.82±0.24 | 87.35±0.17 |
| | GPT* (Radford et al., 2019) | 60.70±0.08 | 78.93±0.17 | 85.09±0.10 | 87.56±0.15 |
| | Center (Wen et al., 2016) | 62.23±0.09 | 79.30±0.14 | 85.20±0.41 | 87.52±0.20 |
| | BSS (Chen et al., 2019) | 60.13±0.32 | 77.98±0.29 | 84.85±0.21 | 87.25±0.07 |
| | **Bi-tuning** | **63.16**±0.26 | **79.98**±0.22 | **86.23**±0.29 | **88.55**±0.38 |

## 5.2 BI-TUNING UNSUPERVISED PRE-TRAINED REPRESENTATIONS

**Bi-tuning representations of MoCo (He et al., 2020).** In this round, we use ResNet-50 pre-trained unsupervisedly via MoCo on ImageNet as the backbone. Since suffering from the large discrepancy between unsupervised pre-trained representations and downstream classification tasks as demonstrated

Table 4: Top-1 Accuracy on **Car** dataset with different *unsupervisedly pre-trained* representations.

| Pre-training Method | Fine-tuning (100% data) | **Bi-tuning** (100% data) |
|---|---|---|
| Deep Cluster (Caron et al., 2018) | 83.90±0.48 | **87.71**±0.34 |
| InsDisc (Wu et al., 2018) | 86.59±0.22 | **89.54**±0.25 |
| CMC (Tian et al., 2019) | 86.71±0.62 | **88.35**±0.44 |
| MoCov2 (He et al., 2020) | 90.15±0.48 | 90.79±0.34 |
| SimCLR(1×) (Chen et al., 2020) | 89.30±0.18 | **90.84**±0.22 |
| SimCLR(2×) (Chen et al., 2020) | 91.22±0.19 | **91.93**±0.19 |

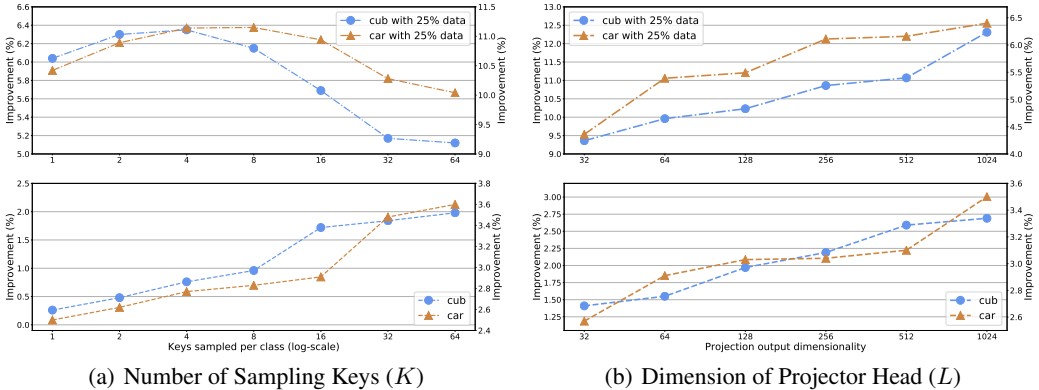

(a) Number of Sampling Keys ($K$)  (b) Dimension of Projector Head ($L$)

Figure 2: Sensitivity analysis of hyper-parameters $K$ and $L$ for Bi-tuning.

in Figure 3, previous fine-tuning competitors usually perform very poorly. Hence we only compare *Bi-tuning* to the state-of-the-art method *BSS* (Chen et al., 2019) and vanilla fine-tuning as baselines. Besides, we add two intuitively related baselines: (1) **GPT\***, which follows a GPT (Radford & Sutskever, 2018; Radford et al., 2019) fine-tuning style but replaces its predictive loss with the contrastive loss; (2) **Center** loss, which introduces compactness of intra-class variations (Wen et al., 2016) that is effective in recognition tasks. As reported in Table 3, trivially borrowing fine-tuning strategy in GPT (Radford & Sutskever, 2018) or center loss brings tiny benefits, and is even harmful on some datasets, *e.g.* **CUB**. Bi-tuning yields consistent gains on all fine-tuning tasks of unsupervised representations, indicating that Bi-tuning benefits substantially from exploring the intrinsic structure.

**Bi-tuning other unsupervised pre-trained representations.** To justify Bi-tuning's general efficacy, we extend our method to unsupervised representations by other pre-training methods. Bi-tuning is applied to MoCo (version 2) (He et al., 2020), SimCLR (Chen et al., 2020), InsDisc (Wu et al., 2018), Deep Cluster (Caron et al., 2018), CMC (Tian et al., 2019) on **Car** dataset with 100% training data. Table 4 is a strong signal that Bi-tuning is not bound to specific pre-training pretext tasks.

**Analysis on components of contrastive learning.** Recent advances in contrastive learning, *i.e.* momentum contrast (He et al., 2020) and memory bank (Wu et al., 2018) can be plugged into Bi-tuning smoothly to achieve similar performance and the detailed discussions are deferred to *Appendix*. Previous works (He et al., 2020; Chen et al., 2020) reveal that a large amount of contrasts is crucial to contrastive learning. In Figure 2(a), we report the sensitivity of the numbers of sampling keys in Bi-tuning (MoCo) under 25% and 100% sampling ratio configurations. Figure 2(a) shows that though a larger key pool is beneficial, we cannot expand the key pool due to the limit of training data, which may lose sampling stochasticity during training. This result suggests that there is a trade-off between stochasticity and a large number of keys. Chen et al. (2020) pointed out that the dimension of the projector also has a big impact. The sensitivity of the dimension of projector head is also presented in Figure 2(b). Note that the unsupervised pre-trained model (*e.g.*, MoCo) may provide an off-the-shelf projector, fine-tuning or re-initializing it is almost the same (90.88 vs. 90.78 on **Car** when $L$ is 128).

**Interpretable visualization of learned representations.** As visualized by Fong & Vedaldi (2017) shown in Figure 3. Note that 3(a) is the original image, Figure 3(b), Figure 3(c) and Figure 3(d) are respectively obtained from a randomly initialized model, a supervised pre-trained model on

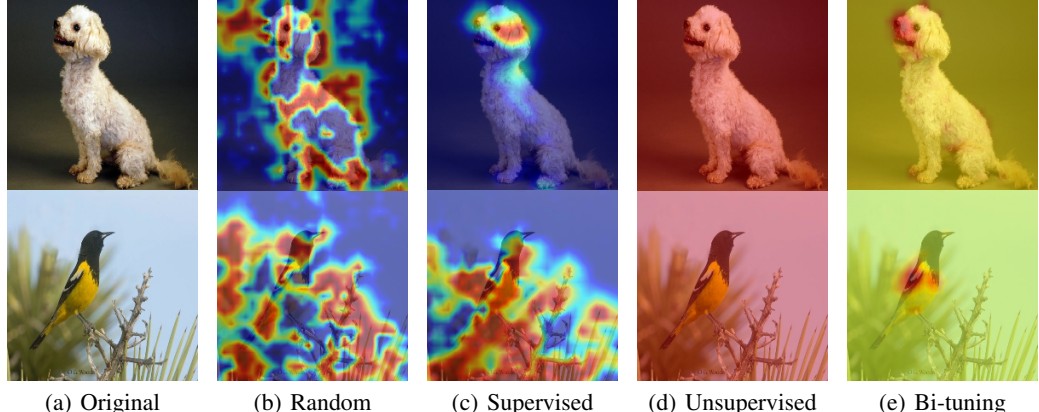

|     (a) Original     |     (b) Random     |     (c) Supervised     |     (d) Unsupervised     |     (e) Bi-tuning     |

Figure 3: Interpretable visualization of learned representations via various training methods.

ImageNet, and an unsupervised pre-trained model via MoCov1 (He et al., 2020). We infer that supervised pre-training will obtain representations focusing on the discriminative part and ignoring the background part. In contrast, unsupervised pre-training pays uninformative attention to every location of an input. Bi-tuning in 3(e) captures both local details and global category-structures.

### 5.3 Collaborative Effect of Loss Functions

Using either contrastive cross-entropy (CCE) or categorical contrastive (CCL) with vanilla cross-entropy (CE) already achieves relatively good results, as shown in Table 5. These experiments prove that there is collaborative effect between CCE and CCL loss empirically. It is worth mentioning that CCE and CCL can work independently of CE (see the fourth row in Table 5), while we optimize these three losses simultaneously to yield the best result. As discussed in prior sections, we hypothesize that Bi-tuning helps fine-tuning models characterize the intrinsic structure of training data when using CCE and CCL simultaneously.

Table 5: Collaborative effect in Bi-tuning on CUB-200-2011 using ResNet-50 pre-trained by MoCo.

| Loss Function | | | Sample Rate | | | |
|:---:|:---:|:---:|:---:|:---:|:---:|:---:|
| CE | CCE | CCL | 25% | 50% | 75% | 100% |
| ✓ | ✗ | ✗ | 38.57±0.13 | 58.97±0.16 | 69.55±0.18 | 74.35±0.18 |
| ✓ | ✓ | ✗ | 45.42±0.11 | 64.33±0.28 | 71.56±0.30 | 75.82±0.21 |
| ✓ | ✗ | ✓ | 41.09±0.23 | 60.77±0.31 | 70.30±0.29 | 75.30±0.20 |
| ✗ | ✓ | ✓ | 47.70±0.41 | 64.77±0.15 | 71.69±0.11 | 76.54±0.24 |
| ✓ | ✓ | ✓ | **50.54**±0.23 | **66.88**±0.13 | **74.27**±0.05 | **77.12**±0.23 |

Table 6: Comparison of different multi-positives contrastive losses on 100% data CUB.

| Losses | Fine-tuning baseline | Bi-tuning with $L'_{\text{InfoNCE}}$ | Bi-tuning with $L_{\text{proposed}}$ |
|:---:|:---:|:---:|:---:|
| Accuracy | 80.74 | 81.20 | 82.93 |

### 6 Conclusion

In this paper, we propose a general Bi-tuning approach to fine-tuning both supervised and unsupervised representations. Bi-tuning generalizes the standard fine-tuning with an encoder for pre-trained representations, a classifier head and a projector head for exploring both the discriminative knowledge of labels and the intrinsic structure of data, which are trained end-to-end by two novel loss functions. Bi-tuning yields state-of-the-art results for fine-tuning tasks on both supervised and unsupervised pre-trained models by large margins. Code will be released upon publication at `http://github.com`.

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

## A    VISUALIZATION BY T-SNE

We train the t-SNE (Maaten & Hinton, 2008) visualization model on the MoCo representations fine-tuned on Pets dataset Parkhi et al. (2012). Visualization of the validation set is shown in Figure 4. Note that representations in Figure 4(a) do not present good classification structures. Figure 4(c) suggests that forcefully combining the unsupervised contrastive learning loss as GPT (Radford et al., 2019) may cause conflict with CE and clutter the classification boundaries. Figure 4(d) suggests Bi-tuning encourages the fine-tuning model to learn better intrinsic structure besides the label information. Therefore, Bi-tuning presents the best classification boundaries as well as intrinsic structures.

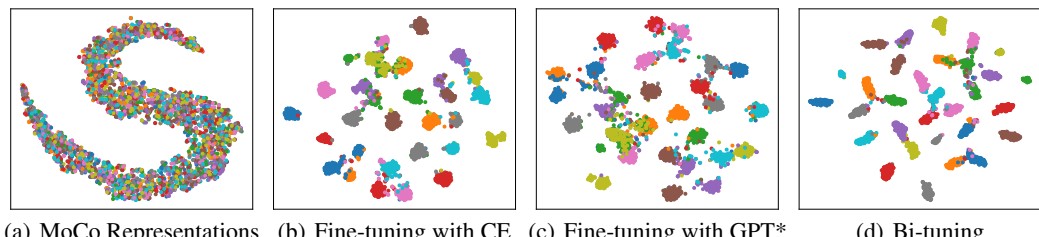

(a) MoCo Representations    (b) Fine-tuning with CE    (c) Fine-tuning with GPT*    (d) Bi-tuning

Figure 4: T-SNE (Maaten & Hinton, 2008) visualization of baselines on Pets (Parkhi et al., 2012).

## B    KEY GENERATING MECHANISMS

### B.1    MOMENTUM CONTRAST

Momentum Contrast (MoCo) (He et al., 2020) is a general key generating mechanism for using contrastive loss. The main idea in MoCo is producing encoded keys on-the-fly via a momentum-updated encoder and maintaining a queue to support sampling operations. Thus, the memory cost in MoCo does not depend on the size of the training set (while a memory bank Wu et al. (2018) will store the whole dataset). In all our experiments (Section 5), Bi-tuning chooses the unsupervised MoCo as our default setting.

Formally, denoting the momentum-updated encoder as $f_k$ with parameters $\theta_k$. Likewise, denoting the backbone encoder as $f_q$ with parameters $\theta_q$. $\theta_k$ is updated by:

$$\theta_k \leftarrow m\theta_k + (1-m)\theta_q. \tag{8}$$

Here we set the momentum coefficient $m = 0.999$. To fit the Bi-tuning approach, we reorganize the queues in MoCo for items in each category separately. Moreover, two contrastive mechanisms in Bi-tuning are performed on different the instance level and category level respectively, and we maintain two groups of queues correspondingly.

Table 7: Top-1 accuracy (%) of Bi-tuning on CUB with memory bank as key generating mechanism (Backbone: ResNet-50 pretrained via MoCo).

| Key Generating Mechanism | Sample Rate | | | |
|---|---|---|---|---|
| | 25% | 50% | 75% | 100% |
| MoCo (He et al., 2020) | 49.25±0.23 | 66.88±0.13 | 74.27±0.05 | 77.12±0.23 |
| Memory bank (Wu et al., 2018) | 50.01±0.55 | 66.69±0.26 | 74.22±0.31 | 77.62±0.29 |

### B.2    MEMORY BANK

Bi-tuning is a general approach, which is not bound to any special key generating mechanism (MoCo). The memory bank proposed by Wu et al. (2018) generates encoded keys via momentum-updated snapshots of all items in the training set. Keys for each mini-batch are uniformly sampled from the memory bank. Compared to MoCo, maintaining a memory bank is more computation-efficient with

more memory required. Similar to Eq. (8), snapshots here are updated by:

$$\mathbf{z}_i^k \leftarrow m\mathbf{z}_i^k + (1-m)\mathbf{z}_i^q, \tag{9}$$

$$\mathbf{h}_i^k \leftarrow m\mathbf{h}_i^k + (1-m)\mathbf{h}_i^q. \tag{10}$$

Notations follow Section 3. Here we set the momentum coefficient $m = 0.5$ (Wu et al., 2018). Other hyper-parameters are the same as Section 5. We evaluate Bi-tuning with a memory bank on CUB (Welinder et al., 2010) with the same configurations in Section 4. The results in Table 7 show that the performance is close in both methods. Key generating mechanisms in Bi-tuning only have limited effects on the final performance in the supervised paradigm. These suggest that the key generating mechanism in Bi-tuning can be implemented by some variants with similar performance. MoCo is recommended regarding its scalability and simplicity.

## C  ADDITIONAL EXPERIMENTAL RESULTS

### C.1  BI-TUNING SUPERVISED PRE-TRAINED REPRESENTATIONS ON MORE BENCHMARK

In addition to the Table 1 in the main paper, we further conduct experiments on Stanford Dogs (Khosla et al., 2011), Oxford-IIIT Pets (Parkhi et al., 2012), Oxford 102 Flowers (Nilsback & Zisserman, 2008), NABirds (Van Horn et al., 2015). Results are shown in Table 8.

Table 8: Top-1 accuracy on more benchmark using ResNet-50 by *supervised pre-training*.

| Dataset | Method | Sampling Rates | | | |
| | | 25% | 50% | 75% | 100% |
| --- | --- | --- | --- | --- | --- |
| Dog | Fine-tuning (baseline) | 86.97 | 88.47 | 88.84 | 89.44 |
| | **Bi-tuning** | **87.07** | **88.72** | **88.99** | **89.52** |
| Pets | Fine-tuning (baseline) | 90.92 | 91.99 | 92.97 | 93.68 |
| | **Bi-tuning** | **90.92** | **92.29** | **93.32** | **93.83** |
| Flower | Fine-tuning (baseline) | 84.84 | 92.50 | 94.91 | 96.08 |
| | **Bi-tuning** | **86.55** | **94.18** | **96.21** | **97.12** |
| NABird | Fine-tuning (baseline) | 38.86 | 61.33 | 69.92 | 73.44 |
| | **Bi-tuning** | **48.33** | **65.81** | **72.36** | **74.83** |

### C.2  BI-TUNING UNSUPERVISED PRE-TRAINED REPRESENTATIONS ON MORE BENCHMARK

Similarly, we also conduct experiment with ResNet-50 unsupervised pretrained by MoCo on Stanford Dogs (Khosla et al., 2011), Oxford-IIIT Pets (Parkhi et al., 2012), Oxford 102 Flowers (Nilsback & Zisserman, 2008), NABirds (Van Horn et al., 2015). Results are shown in Table 9 as a suppmentary to Table 3 to report results on more datasets using ResNet-50 *unsupervisedly pre-trained* by MoCo. Further, to clarify that Bi-tuning is not overfitting to pre-training methods, Table 10 and Table 11 provide more promising results with different *unsupervisedly pre-trained* representations on **CUB** and **Aircraft** respectively, which reveal that Bi-Tuning consistly outperforms fine-tuning method.

Table 9: Top-1 accuracy on more benchmark using ResNet-50 *unsupervised pre-trained* by MoCo.

| Dataset | Method | Sampling Rates | | | |
| | | 25% | 50% | 75% | 100% |
|---|---|---|---|---|---|
| Dog | Fine-tuning (baseline) | 54.95 | 68.38 | 72.51 | 75.57 |
| | **Bi-tuning** | **59.32** | **69.52** | **73.25** | **76.65** |
| Pets | Fine-tuning (baseline) | 70.89 | 80.68 | 84.40 | 86.30 |
| | **Bi-tuning** | **74.14** | **82.94** | **86.09** | **87.79** |
| Flower | Fine-tuning (baseline) | 77.96 | 89.69 | 92.91 | 95.25 |
| | **Bi-tuning** | **81.01** | **90.70** | **94.13** | **95.43** |
| NABird | Fine-tuning (baseline) | 56.36 | 69.75 | 74.85 | 77.10 |
| | **Bi-tuning** | **62.11** | **73.26** | **77.29** | **79.50** |

Table 10: Top-1 Accuracy on **CUB** dataset with different *unsupervisedly pre-trained* representations.

| Pre-training Method | Fine-tuning (100% data) | **Bi-tuning** (100% data) |
|---|---|---|
| Deep Cluster (Caron et al., 2018) | 74.63 | **78.24** |
| InsDisc (Wu et al., 2018) | 71.35 | **74.92** |
| CMC (Tian et al., 2019) | 68.71 | **77.21** |
| MoCov2 (He et al., 2020) | 75.75 | **77.24** |
| SimCLR(1×) (Chen et al., 2020) | 72.21 | **77.44** |
| SimCLR(2×) (Chen et al., 2020) | 74.35 | **78.43** |

Table 11: Top-1 Accuracy on **Aircraft** dataset with different *unsupervisedly pre-trained* representations.

| Pre-training Method | Fine-tuning (100% data) | **Bi-tuning** (100% data) |
|---|---|---|
| Deep Cluster (Caron et al., 2018) | 77.68 | **81.70** |
| InsDisc (Wu et al., 2018) | 84.16 | **86.08** |
| CMC (Tian et al., 2019) | 83.98 | **85.84** |
| MoCov2 (He et al., 2020) | 88.04 | **89.85** |
| SimCLR(1×) (Chen et al., 2020) | 86.52 | **89.10** |
| SimCLR(2×) (Chen et al., 2020) | 88.39 | **89.98** |

