# OpenReview forum: "Bi-tuning of Pre-trained Representations"
_ICLR.cc/2021/Conference — Reject_

### Official Review · AnonReviewer2 · 2020-10-23
**Official Blind Review #2**

**Rating:** 4
**Confidence:** 5

**Review:**

This paper proposes a general Bi-tuning approach to fine-tuning both supervised and unsupervised representations to downstream tasks. The main contribution of this paper is integrating two heads upon the backbone of pre-trained representations: a classifier head with an improved contrastive cross-entropy loss and a projector head with a newly-designed categorical contrastive learning loss. Results are shown that Bi-tuning achieves state-of-the-art results for fine-tuning tasks of both supervised and unsupervised pre-trained models by large margins.

Pros:

+ Overall, the paper is well written. In particular, the Introduction section has a nice flow and puts the proposed method into context. Despite the method having limited novelty (only improve the loss function on the basis of contrastive learning), the method has been well-motivated by pointing out the limitations in SOTA methods.

+ The idea of designing a novel categorical contrastive loss $L_{CCL}$ on the projector head, by expanding the scope of positive keys to a set of instances instead of a single one, it is very interesting.

+ The results section is well structured. It's nice to see analysis on components of contrastive learning and the collaborative effect of the loss functions.


Cons:

- The key concern about the paper is the lack of enough novelty. It seems that the core idea of Bi-tuning is the last overall loss function: $L_{CE} + L_{CCE} + L_{CCL}$, but the author stated that "the naive combination of the supervised cross-entropy loss and the unsupervised contrastive loss is not an optimal solution for fine-tuning" in section 4.4. In my opinion, $L_{CE}$ and $L_{CCE}$ are from the classifier head, they are supervised cross-entropy loss, while $L_{CCL}$ is from the projector head, it is an unsupervised contrastive loss. Isn't adding these three values a "naive combination"? If the core idea is only improved the loss function, I think such novelty is limited.

- Considering the comprehensive experiments, a deeper analysis of the proposed method would have been nice. The idea of novel categorical contrastive loss $L_{CCL}$ on the projector head, expanding the scope of positive keys to a set of instances, what kind of scope? What effect will the scope change have on the result? The author stated that "resulting in more harmonious cooperation between the supervised and unsupervised learning mechanisms", some form of theoretical/analytical reasoning behind the effectiveness would provide greater insights into the community and facilitate further research in this direction.

- The claim of the 10.7\% absolute rise on CUB with a sampling rate of 25\% mentioned in the abstract has not been clearly addressed and pointed out in the results section of the paper. From Table 1, I don't know how to calculate the 10.7\% **absolute rise**. Do you actually mean the **relative rise** including *the margin of error*?

- Fig 3: Why interpretable? The author stated that "The visualization shows that MoCo focuses only on local details.", I don't understand how Fig 3(d) expresses local features. What's more, Fig 3 was quoted many times in the paper, such as sections 2.3 and 5.2, but I really don't understand the connection with Fig 3.

- In section 5.3, the author stated "It is worth mentioning that CCE and CCL can work independently of CE", why not set up experiments for CCE and CCL is independent?


Minor comments:

* In Section 5.2, "As reported in Figure 3", the link of Fig 3 is an error.

* The labels of the figures and tables in the paper should preferably appear and quote in order.

---

> ### Author Response · Authors · 2020-11-21
> **Response to AnonReviewer2**
>
> Thanks for your insightful review. We will address your comments through both feedback and revision.
>
> **Q: Novelty and naive combination of the losses:**
>
> - The statement "The naive combination of the supervised cross-entropy loss and the unsupervised contrastive loss is not an optimal solution for fine-tuning" is referred to the widely-used GPT-style combination: jointly optimizing the loss used in pre-training stage (unsupervised contrastive loss here) and the loss of downstream task (cross-entropy for classification).
> - In the experiment section, we have evaluated that straightforward combination of contrastive loss + cross-entropy for fine-tuning. As shown in the **GPT\*** row of Table 3, the inclusion of  "plain" contrastive loss is unsatisfactory. The proposed Bi-tuning outperforms the "plain" method by $3.90\%$ on average.
> - Further, as shown in Figure 4c of the appendix,  the tSNE visualization of fine-tuning with **GPT*** is less discriminative than that of Bi-tuning, which indicates intuitively that the "plain" contrastive loss conflicts with the standard cross-entropy loss.
> - Specifically, the "plain" contrastive loss will push each data point apart from the others, while the standard cross-entropy loss will pull close the data points of the same class. Conflict happens. By seamlessly incorporating the class information into the proposed losses, such a conflict is readily resolved.
> - This paper embeds the idea of contrastive learning into the fine-tuning stage in a dual-head architecture and applies two newly designed contrast-based losses that tailor better to downstream classification tasks. In particular, $L_\text{CCE}$ is a novel idea to integrate contrast into cross-entropy loss.
>
>
> **Q: What is the scope you apply contrasts**
>
> - We expend Section 3, providing more insights into our proposed losses. Here we further clarify that, besides the contrasts constructed from data augmentations, we further expand the scope of positive keys using all the instances from the same class to the query.
>
>
> **Q: 10.7% absolute rise**
>
> - This statement refers to Table 3. When fine-tuning MoCo (ResNet-50) to CUB with 25% data, we achieve 10.7% absolute rise. We will clarify these misleading statements in the revision.
>
> **Q: Visualization in Figure 3**
>
> - Thanks for your detailed reviews; we fix some wrong links to Figure 3.
>
> - Figure 3 is a set of attention maps attained by a visualization tool which shows discriminative localization (Fong et al., 2017). We infer that supervised pre-training will obtain representations focusing on the discriminative part and ignoring the background part as shown in Figure 3(c). In contrast, unsupervised pre-training pays uninformative attention to every location of an input as shown in Figure 3(d). We hypothesize that this phenomenon may contribute to effective performance when fine-tuning to downstream tasks that focus more on fine-grained local information. And the attention maps of Bi-tuning in Figure 3(e) indicate that fine-tuning to classification task is more similar to supervised pre-training than to unsupervised pre-training.
>
> **Q: Set up experiments for CCE and CCL independently**
>
> - Our ablations reveal that using CCE and CCL simultaneously without CE can obtain a comparable result. We cannot use CCL alone because it is only performed on the projector head and does not directly supervise the classifier. CCE can work in place of the conventional CE loss but shows no obvious superiority over CE. So we tend to recommend using CCE and CCL simultaneously to boost CE loss.

---

### Official Review · AnonReviewer3 · 2020-10-27
**good empirical results, unclear intuition and justification**

**Rating:** 4
**Confidence:** 4

**Review:**

Summary:
The authors propose to augment the regular fine-tuning stage by two additional loss: a contrastive cross-entropy loss (L_CCE) and a categorical contrastive learning loss (L_CCL).  For each input example, L_CCE minimizes the relative distance between its representation and the softmax linear weight (compared to the distance between another image from the same class and the linear weight). L_CCl encourages representations from the same class to stay close with each other. Empirical results on CUB, Cars, Aircraft and COCO-70 show the advantage of the method.

Strength:
1. Exploring the intrinsic structure of the downstream task is an interesting direction to improve transfer learning performance.
2. The proposed method shows consistent improvements across various datasets.

Weakness:
The additional loss terms are not well-motivated and difficulty to justify. How do they help the model exploit the intrinsic data structure?
  a. In Eq3, L_CCE computes the log ratio between
        {the distance between the representation h_{i}^{q} and the weight of the true class w_{y_i}}
      and
        the sum of {the distance between the positive key set h_{j}^{k} and the weight of the true class w_{y_i}}
  What does this log ratio term represent? The numerator can be bigger than the denominator in this case (and then the log term will be negative). Also if you just want to encourage instances from the same class to stay closer in the representation space, why don't you just explicitly minimize the L2 distance between the representation and the prototype?
  b. The goal of L_CCL is to encourage examples from the same class to stay closer with each other. Since the positive and negative sets are constructed using the label information, I don't see how it is complimentary to the original cross entropy loss. Also, in the case where the size of the positive set |S_i| is bigger than one, equation 4 may not make much sense as there are positive examples in the denominator.

Maybe I missed or misunderstood something, but based on these questions and concerns, I found it hard to justify the improvement.

---

> ### Author Response · Authors · 2020-11-21
> **Response to AnonReviewer3**
>
>
> Thanks for your detailed comments. We reorganize Section 4 in our new revision. We also rewrite the formulas in a more readable manner to clarify any possible confusion.
>
> **Q**: Why there are positive examples contributing to the denominator in $L\_\text{CCL}$ and $L\_\text{CCE}$:
>
> The straightforward approach to expand the standard contrastive loss ($L\_{\text{InfoNCE}}$) into a form of multi-positives (denoted by a positive key set ${|\mathbf{K}\_p|}$) is
>
> $$
> {L'}\_\text{InfoNCE}=-\frac{1}{|\mathbf{K}\_p|}\sum\_{\mathbf{k}\_{+}\in \mathbf{K}\_p}\log\frac{\exp(\mathbf{q}\cdot \mathbf{k}\_{+}/\tau)}{\exp(\mathbf{q}\cdot \mathbf{k}\_{+}/\tau)+\sum\_{\mathbf{k}\_{-}\in {\mathbf{K}\_n}} \exp(\mathbf{q}\cdot \mathbf{k}\_{-}/\tau)}
> $$
>
> where ${|\mathbf{K}\_n|}$ denotes the negative key set. We denote this loss function by ${L'}\_\text{InfoNCE}$, which essentially performs multiple **individual** contrasts with different positive keys for each query $q$. The losses ($L\_\text{CCL}$ and $L\_\text{CCE}$) we proposed are based on the following formula:
>
> $$
> L\_{\text{proposed}}=-\frac{1}{|\mathbf K\_p|}\sum\_{\mathbf{k}\_{+}\in \mathbf K\_p}\log\frac{\exp(\mathbf{q}\cdot \mathbf{k}\_{+}/\tau)}{\sum\_{k^+\in \mathbf K\_p}\exp(\mathbf{q}\cdot \mathbf{k}\_{+}/\tau)+\sum\_{ \mathbf{k}\_{-}\in {\mathbf K\_n}} \exp(\mathbf{q}\cdot \mathbf{k}\_{-}/\tau)}
> $$
>
> Different from ${L'}\_\text{InfoNCE}$, in the denominator of $L\_{\text{proposed}}$, both the positive keys in the same class with the query and the negative keys from other classes are presented. For each contrast with multiple positive keys, a query here needs to *balance* all positive keys simultaneously. In another view, $L\_{\text{proposed}}$ can be regarded as performing cross-entropy on soft labels of uniform probability $\frac{1}{|\mathbf K\_p|}$ for each positive key. $L\_\text{proposed}$ introduces a *uninformative prior* that the positive keys are uniformly distributed around the query.
>
> We discuss these two forms (${L'}\_\text{InfoNCE}$ and $L\_{\text{proposed}}$) in the revision and conduct them on CUB datasets with $100$% data. $L\_\text{proposed}$ yields more accuracy gains than ${L'}\_\text{InfoNCE}$.
>
> | CUB 100% | Fine-tuning | Bi-tuning with ${L'}_\text{InfoNCE}$ | Bi-tuning with $L_{\text{proposed}}$ |
> | -------- | ----------- | ------------------------------------- | ------------------------------------ |
> | Accuracy | 80.74       | 81.20                                 | 82.93                                |
>
> **Q: How is $L\_{\text{CCL}}$ complimentary to the original $L\_\text{CE}$:**
>
> The two losses have the following fundamental distinction:
>
> - Through standard cross-entropy loss $L\_\text{CE}$, we will learn a hyperplane for discriminating each class from the other classes, and the instances of each class are only required to be far away from its associated hyperplane---they are not required to form into a compact structure in the metric space.
>
> - Through the proposed categorical contrastive loss $L\_{\text{CCL}}$, besides requiring the instances of each class to stay far away from those of the other classes, we further require that they should form a compact structure in the metric space where each is close to the other. This is exactly the advantage of *contrast-by-metric* over *far-from-hyperplane*.
>
> - Hence, the proposed $L\_{\text{CCL}}$ can be complimentary to the original $L\_\text{CE}$.
>
>
> **Q: $L\_{\text{CCE}}$ clarification:**
>
> - Except for the way of computing the logits, $L\_\text{CCE}$ has similar form as $L\_\text{proposed}$. As the numerator term also appears in the denominator, the fraction will be never larger than $1$. Overall $L\_\text{CCE}$ is an interesting hybrid of $L\_\text{CE}$ and $L\_\text{InfoNCE}$. The log term is introduced due to the cross-entropy over the softmax-based probability.
> - $L\_\text{CCE}$ (Eq. 3) can be regarded as a dual view of the conventional cross-entropy loss $L\_{\text{CE}}$ (Eq. 2). As revealed in Figure 1, to find the correct class, $L\_\text{CCE}$ performs column-wise championship while $L\_{\text{CE}}$ performs row-wise championship. Further, instead of operating loss computation along the class dimension (*i.e.* number of classes),  $L\_\text{CCE}$ operates along the key-set dimension ($K+1$). Further, if we regard $L\_{\text{CE}}$ as a class-wise championship, then $L\_\text{CCE}$ can be regarded as an instance-wise championship.
>
> - Experiments in Table 3 show that minimizing the $L\_2$ distance between the representation space and the class prototype only brings limited benefits, as shown in the **Center** (a.k.a. the center loss which assumes Gaussian for each class) row of the table. Bi-tuning outperforms these competitors significantly.

---

### Official Review · AnonReviewer4 · 2020-10-28
**Using contrastive losses may improve fine-tuning**

**Rating:** 5
**Confidence:** 3

**Review:**

### Summary

The paper suggests to extend fine-tuning of pre-trained representations by adding additional contrastive losses to leverage the intrinsic structure of the downstream training data. The authors call the presented method Bi-tuning and evaluate it by comparing it to other variants of fine-tuning.

### Quality, clarity, originality and significance

The paper describes its motivation well, and Sections 1 and 2 are well-written and clear. Sections 3, 4, and especially 5, I found less clear and in some parts a bit confusing.

One point that I found not entirely clear in the motivation is the focus on the conjunction of *fine-tuning* and *supervised contrastive learning*. Others have looked at the latter for general training, e.g. [arXiv:2004.11362](https://arxiv.org/pdf/2004.11362.pdf) or [arXiv:1808.04699](https://arxiv.org/pdf/1808.04699.pdf), so it is not clear to me why the Bi-tuning loss is restricted (in this study) to fine-tuning only. If the loss is universally applicable, shouldn't it also work for training in general? Or if there is a reason to not expect this (i.e. to expect that it should only work well with pre-training) this should be explained (or the explanation emphasized if I missed it). If the focus is on bringing contrastive learning to the fine-tuning stage, then maybe an inclusion of some more "plain" contrastive  losses on the way to the "novel" loss introduced here may lead to a better understanding, focusing specifically on the fine-tuning stage.

The experimental results look good at first glance, but overall I found it hard to evaluate how convincing they are because of several potential problems:
* The experimental results are not compared to any results from the existing literature, all baselines and comparisons are entirely from this paper. This makes it hard to know how much careful tuning went into the proposed algorithm versus the baselines. I think at least *some* comparison to results from the literature should be possible or it should be carefully explained why such an overfitting to the proposed algorithm clearly did not happen. For example, the [SpotTune paper](https://www.cs.utexas.edu/~grauman/papers/CVPR19_spottune.pdf) contains numbers for CUBS and Cars that look comparable at first glance because they also use pre-trained ResNet-50 models, and they include numbers for L2-SP that are better than the ones in this paper. It may be very insightful to compare results to these or other similar results. There seems to be some overlap with methods/datasets to (Chen et al., 2019), but then again there are differences like the percentages of data samples.
* There are a lot of seemingly random choices in the selection of the datasets, e.g. why CUB/Cars/Aircraft and not e.g. CIFAR, Dogs, Pets, Flowers, or Birds? I'm not saying any of these choices are inherently better, but the authors should explain *why* these datasets are chosen, to avoid the impression that the datasets may have been chosen because the proposed method works particularly well on these dataset combinations. One way to make the results stronger would therefore be to use combinations of datasets that other, previous work has already used, which would also enable a direct comparison (see above).
* Again in a similar fashion, Table 4 evaluates only one target data set, why only this one?

### Pros and cons
* Pros: interesting exploration of fine-tuning with added contrastive losses
* Cons: experimental results not very convincing, main message is not very clear

### Minor details and comments
* Novelty of the presented components seemed a bit overstated to me in some parts. E.g. words like "improved", "novel", "newly-designed" are used throughout the work when it is not entirely clear (at least to me) where exactly the novelty lies.
* " the best learning rate is selected by cross-validation under a 100% sampling rate and applied to all four sampling rates." - how exactly is the cross-validation performed here?
* Sec. 5.1. does not seem to specify the pretraining dataset. I assume it's ImageNet, but giving some specifics (e.g. was a particular publicly available checkpoint used) might be helpful.
* "Previous fine-tuning methods mainly focus on improving performance under low-data regime paradigms." It seems recently there are several works using fine-tuning to ImageNet, which may count as large-scale, too? E.g. [PapersWithCode-ImageNet](https://paperswithcode.com/sota/image-classification-on-imagenet) has several approaches near the top-performing methods that would fall into that category I think.
* Figure 2(b): performance seems not yet saturated going to higher dimensionality - would it make sense to go even higher?
* Figure 3 is very hard to interpret I think and it is only explained extremely briefly by citing a reference. Why is the visualization uniform in case (d)? What does the visualization show? Which model and data were used? Also the conclusion "Bi-tuning in 3(e) captures both local details and global category-structures" is not evident from the figure alone, I think it may need more explanation and interpretation.

---

> ### Author Response · Authors · 2020-11-18
> **Response to AnonReviewer4 (Part I) [1/2]**
>
> We appreciate your detailed comments. We have conducted more experiments following your suggestions and appended them to the revision.
>
> **Q: The focus of this paper**
>
> - We clarify that this paper focuses on the fine-tuning stage since fine-tuning is one of the most classical and popular paradigms of transfer learning.
>
> - As for pre-training, the proposed contrastive losses cannot apply to unsupervised pre-training due to the requirement of labeled data. We agree with the reviewer that, while out of the scope of the current paper, exploring the proposed losses in the supervised pre-training stage or general training is a promising future research direction.
>
> - In the experiment section, we have evaluated that straightforward combination of contrastive loss with fine-tuning, i.e. the inclusion of  "plain" contrastive loss as mentioned by the reviewer,  is unsatisfactory (see the *GPT** row in Table 3). The proposed Bi-tuning outperforms the "plain" method by 3.90% on average.
>
> - Further, as shown in Figure 4c of the appendix,  tSNE visualization of the fine-tuning with GPT* is less discriminative than that of Bi-tuning, which indicates that the 'plain' contrastive loss intuitively conflicts with the standard cross-entropy loss.
> - Specifically, the "plain" contrastive loss will push each data point apart from the others (*i.e.*, even points of the same class should be drawn far apart), while the standard cross-entropy loss will pull data points of the same class to be on the same side of its hyperplane. Conflict happens because we will come up with large dispersion within each class. By seamlessly incorporating the class information into the proposed losses, such a conflict is readily resolved.
>
> **Q: Why only compared to the baselines reproduced by this paper.**
>
> -  We mainly follow the experiment settings in (Chen et al., 2019), which has released a complete code.
>
> - There are two reasons that we cannot directly compare the results to *SpotTune* (Guo et al., 2018):
>
>   1. Different experiment settings and implementation protocols:
>      - *SpotTune* uses the fine-tuned network as well as the pre-trained network for inference, which implies that the volume of  parameters is doubled. This is not common since in practice, we usually use the fine-tuned network for inference.
>      - *SpotTune* uses the protocol of ten-crop prediction in the testing phase for all of its compared methods, leading to a generally higher accuracy.
>
>   2. Missing details for evaluating *SpotTune* on our experiment settings:
>      - *SpotTune* applies different data augmentations for different datasets without explaining the implementation details. Also, it does not provide the details of the hyper-parameters.
>      - We notice that the authors did not provide a complete code of *SpotTune* (only ResNet-26 on a fraction of the datasets are available), so we cannot produce its results using our experiment settings.
>
> - Compared to (Chen et al., 2019), our evaluation protocols are fair to all competitors.
>
>   1.  All methods are evaluated using exactly the same protocol: PyTorch's builtin data augmentation and one-crop prediction. We believe this protocol is more commonly used, and we do not over-tailor all the methods to these common tricks.
>
>   2. We use the same learning rate strategy for all methods: 5-fold cross-validation on the training set to select learning rate from {$10^{-1},10^{-1.5},10^{-2},10^{-2.5},10^{-3},10^{-3.5},10^{-4},10^{-4.5},10^{-5}$}. This parameter tuning is a standard practice in fine-tuning.
>
> **Q: Detailed explanation of Figure 3**
>
> Figure 3 is a set of attention maps attained by a visualization tool which shows discriminative localization (Fong et al., 2017). We infer that supervised pre-training will obtain representations focusing on the discriminative part and ignoring the background part as shown in Figure 3(c). In contrast, unsupervised pre-training pays uninformative attention to every location of an input as shown in Figure 3(d). We hypothesize that this phenomenon may contribute to effective performance when fine-tuning to downstream tasks that focus more on fine-grained local information. And the attention maps of Bi-tuning in Figure 3(e) indicate that fine-tuning to classification task is more similar to supervised pre-training than to unsupervised pre-training.

---

> > ### Comment · AnonReviewer4 · 2020-11-24
> > **Thank you for your response**
> >
> > Thank you for your detailed response and for revising the paper. I have read your response and will take it into account in the discussion period.

---

> ### Author Response · Authors · 2020-11-18
> **Response to AnonReviewer4 (Part II) [2/2]**
>
> **Q: The chosen sample ratio configurations and datasets.**
>
> - In this paper, we modify the original sampling ratios from ($15$%, $30$%, $50$%,$100$%) to ($25$%, $50$%, $75$%,$100$%). The reason is that on some of the datasets, $15$% of training data is too small for fine-tuning from the pre-trained models, e.g. $15$% of CUB means only 899 samples for 200 greeds.
>
> - To clear the concern, we conduct experiments on $15$% of dataset, and confirm that Bi-tuning brings more significant gains in such a low-sample ratio. Results are shown below (ResNet-50, based on supervised pre-trained models from ImageNet).
>
> | Dataset      | Fine-tuning | Bi-tuning |
> | ------------ | ----------- | --------- |
> | CUB 15%      | 49.53       | **57.09** |
> | CUB 30%      | 63.46       | **70.18** |
> | Car 15%      | 39.44       | **49.01** |
> | Car 30%      | 64.07       | **72.55** |
> | Aircraft 15% | 38.72       | **47.61** |
> | Aircraft 30% | 56.77       | **64.24** |
>
> - To clear the concern, we also follow datasets in (Chen et al., 2019). We removed two relatively simpler datasets (Dogs and Pets, which are very similar or have large overlap with ImageNet, and also show only marginal effects in previous literature). The results on these four datasets are displayed below (ResNet-50, unsupervised pre-training via MoCo from ImageNet) and Bi-Tuning also works well. In addition, we run another two benchmarks **Bird** and **Flower** in this setting.
>
> | Dog (supervised) | 25%   | 50%   | 75%   | 100%  |
> | ---------------- | ----- | ----- | ----- | ----- |
> | Fine-tuning      | 86.97 | 88.47 | 88.84 | 89.44 |
> | Bi-tuning        | **87.07** | **88.72** |**88.99** |**89.52** |
>
> | Pets (supervised) | 25%   | 50%   | 75%   | 100%  |
> | ----------------- | ----- | ----- | ----- | ----- |
> | Fine-tuning       | 90.92 | 91.99 | 92.97 | 93.68 |
> | Bi-tuning         | **91.77** | **92.29** | **93.32** | **93.83** |
>
> | Flower (supervised) | 25%   | 50%   | 75%   | 100%  |
> | ------------------- | ----- | ----- | ----- | ----- |
> | Fine-tuning         | 84.84 | 92.5  | 94.91 | 96.08 |
> | Bi-tuning           | **86.55** | **94.18** | **96.21** | **97.12** |
>
> | Bird (supervised) | 25%   | 50%   | 75%   | 100%  |
> | ----------------- | ----- | ----- | ----- | ----- |
> | Fine-tuning       | 38.86 | 61.33 | 69.92 | 73.77 |
> | Bi-tuning         | **48.33** | **65.81** | **72.36** | **74.83** |
>
> - To make the results complete, we further conduct experiments on all these four datasets for MoCo.
>
> | Dog (MoCo)  | 25%   | 50%   | 75%   | 100%  |
> | ----------- | ----- | ----- | ----- | ----- |
> | Fine-tuning | 54.95 | 68.38 | 72.51 | 75.57 |
> | Bi-tuning   | **59.32** | **69.52** | **73.25** | **76.65** |
>
> | Pets (MoCo) | 25%   | 50%   | 75%   | 100%  |
> | ----------- | ----- | ----- | ----- | ----- |
> | Fine-tuning | 70.89 | 80.68 | 84.40 | 86.30 |
> | Bi-tuning   | **74.14** | **82.94** | **86.09** | **87.79** |
>
> | Flower (MoCo) | 25%   | 50%   | 75%   | 100%  |
> | ------------- | ----- | ----- | ----- | ----- |
> | Fine-tuning   | 77.96 | 89.69 | 92.91 | 95.25 |
> | Bi-tuning     | **81.01** | **90.70** | **94.13** | **95.43** |
>
> | Bird (MoCo) | 25%   | 50%   | 75%   | 100% |
> | ----------- | ----- | ----- | ----- | ---- |
> | Fine-tuning | 56.36 | 69.75 | 74.85 | 77.1 |
> | Bi-tuning   | **62.11** | **73.26** | **77.29** | **79.5** |
>
> - We agree that Table 4 needs to been expanded with the results of the other two datasets, and we conduct the experiments. The results also demonstrate that Bi-Tuning outperforms the other fine-tuning methods.
>
> | CUB          | Fine-tuning | Bi-tuning |
> | ------------ | ----------- | --------- |
> | Deep Cluster | 74.63       | **78.24**     |
> | InsDisc      | 71.35       | **74.92**     |
> | CMC          | 68.71       | **72.21**     |
> | MoCov2       | 75.75       | **77.24**     |
> | SimCLR (1x)  | 72.21       | **77.44**     |
> | SimCLR (2x)  | 74.35       | **78.43**     |
>
> | Aircraft     | Fine-tuning | Bi-tuning |
> | ------------ | ----------- | --------- |
> | Deep Cluster | 77.68       | **81.70**     |
> | InsDisc      | 84.16       | **86.08**     |
> | CMC          | 83.98       | **85.84**     |
> | MoCov2       | 88.04       | **89.85**     |
> | SimCLR (1x)  | 86.52       | **89.10**     |
> | SimCLR (2x)  | 88.39       | **89.98**     |
>
> - All unsupervised pre-trained weights pickles can be found in their corresponding official sites. For supervised pre-training, we use the pickle in 'torchvision'.

---

### Official Review · AnonReviewer1 · 2020-10-31
**Great work using contrastive losses during fine-tuning**

**Rating:** 8
**Confidence:** 4

**Review:**

== Summary ==

The paper proposes to use a contrastive cross-entropy loss during fine-tuning, for improving transfer accuracy of both supervised and unsupervised pre-training methods in image classification. The paper builds on the intuition that both class discriminative information and intrinsic structure of the downstream task are useful for fine-tuning, and existing fine-tuning approaches only use the former. The authors conduct experiments on four image classification datasets, using a modern ResNet-50 architecture.

== Pros ==

- The authors use two contrastive losses during fine-tuning, which has not been deeply explored, since most works typically use only class cross-entropy loss. One contrastive loss acts on the classification head, while the second, an extension of InfoNCE, acts on a projection head (on top of the representation layer).

- The proposed method achieves very good results across the four image classification datasets (CUB, Cars, Aircraft and a custom version of MS-COCO), using both supervised and unsupervised pre-training. The authors perform 5 runs for each experiment and report the average as well as statistical significance metrics (although it's not clear if the provided interval is standard deviation or a confidence interval).

- When using unsupervised pre-training, the authors explored 6 different pre-training algorithms, and Bi-tuning improves the standard supervised fine-tuning in all cases.

- The authors also show that their method outperforms 4 baselines: standard (supervised) fine-tuning, BSS, DELTA, and L2SP by a significant margin across different data sizes available for fine-tuning (25%, 50%, 75% and 100% of the original dataset size).

- Despite the fact that they have three terms in the final loss, no additional hyper-parameter needs to be introduced (in addition to choosing the number of keys, which depends on the amount of training data).

== Cons ==

- Despite the fact that contrastive losses have not been widely used for fine-tuning (as far as this reviewer is aware), the authors should probably tone-down statements such as "Bi-tuning, a general learning framework to fine-tuning both supervised and unsupervised pre-trained representations to downstream tasks". There's a plethora of works using multiple loss/regularization terms during fine-tuning, such as DELTA or BSS, which the authors compare to. The proposed work only proposes a different type of loss/regularization. One could even argue that "Bi-tuning" is only a particular instance of multi-task learning.

- The authors could have run additional experiments with other modern deep neural network architectures alternative to the ResNet50 (Inception, EfficientNet, DenseNet, AmoebaNet, etc.) to show whether the benefits transfer to other architectures.

- The fact that the authors coined the method "Bi-tuning" but the loss has three terms (CE, CCE and CCL) is confusing. Probably, the "Bi" is due to the two heads, but still.

== Typos ==

The paper contains several typos and some parts are not clear or could be improved, please read it carefully, correct them. Some typos that I've found.

- Not a typo, but you should indicate whether intervals in Tables are +/- std. deviation or confidence intervals.
- Not a typo, but there are much earlier works showing that fine-tuning (usually) performs better than training from-scratch. In the introduction you cite He et al. (2019). A quick search on the Internet yields a survey from 2009 on the topic, for example.
- Equation 1, bold q, k_+ and k_i, since these denote vectors.
- Figure 1, missing h_0^k (according to equation 3, j starts at 0).
- "Previously, we propose" -> "Previously, we proposed".
- "As shown in 1" -> "As shown in Table 1".
- "Results in Table 2 reveal that Bi-tuning brings general gains for all tasks, even provided with sufficient training data". I don't understand the phrase after the comma.
- "SimClR" -> "SimCLR" in Table 4.
- The result of Bi-training with MoCov2 (Table 4) is not statistically significant than the supervised baseline, thus it should not be bolded.

== Reasons for score ==

The authors show that the proposed method achieves significantly better results than strong baselines across multiple datasets, data set sizes, pre-training paradigms (supervised vs. unsupervised) and algorithms. Despite introducing an additional head and two additional losses during fine-tuning, the proposed algorithm does not introduce more hyper-parameters than other alternatives, which makes it easier to be applied by other researchers. This reviewer believes that some claims are a bit overstated (e.g. claiming that the proposed approach is a "framework"), but there's no doubt that the proposed algorithm achieves excellent results and the experimental work is solid.

---

> ### Author Response · Authors · 2020-11-18
> **Response to AnonReviewer1**
>
> We thank you for your encouraging feedback and constructive suggestions.
>
> **Q: Possibly overstate**
>
> We have revised the paper accordingly by replacing 'framework' with 'approach'.
>
> **Q: Additional experiments with other modern deep neural network architectures**
>
> - As a fine-tuning method, Bi-tuning is based on the publicly available pre-trained models, in which ResNet is the most popular architecture and has pre-trained models for different pre-training methods and datasets. So we used ResNet as the main backbone, including ResNet ($1\times$) and ResNet ($2\times$).
>
> - Further, we have already applied our approach to **DenseNet-121** on COCO-70 dataset in Table 2 of the original paper, showing that the benefits of our method are not restricted in ResNet.
>
> **Q: Does 'Bi' due to bi-head?**
>
> Yes. We confirm that the 'Bi' actually refers to the two heads and we have emphasized it in our new revision.
>
> **Q: Writing**
>
> Thanks for your careful reading of our paper. We have fixed the mentioned typos in the revision, which further underwent several rounds of proofreading.

---

### Decision · Program_Chairs · 2021-01-07
**Final Decision**

**Decision:**

Reject

**Comment:**

The authors propose an alternative fine-tuning procedure by introducing a projection head and two new losses to be combined with the vanilla cross-entropy loss. The authors introduce and jointly optimize the standard cross-entropy loss, the contrastive cross-entropy loss for classifier head and the categorical contrastive learning loss for projector head in an end-to-end fashion. The authors empirically confirmed that this setup compares favorably to existing baselines.

The reviewers found the setting challenging and worth investigating. The idea of exploring the intrinsic structure of the downstream task to help with fine-tuning was deemed useful. The reviewers appreciated the thorough empirical validation. While the proposed approach was not yet explored in this specific context, most reviewers were concerned with the lack of novelty. In addition, there seems to be a large gap between the quality of exposition in the introduction and results section with respect to the rest of the paper which introduces confusion.

As it currently stands, the paper is not yet ready for publication and I will recommend rejection. To improve the manuscript the authors should incorporate the received feedback and significantly improve the exposition and justification of the proposed loss. In terms of empirical results, the authors should also explore alternative neural architectures to validate whether the proposed approach is general and whether the need for hyperparameter tuning arises.